# SAVIOR: Sample-efficient Alignment of Vision-Language Models for OCR Representation

## Abstract

Modern enterprises are increasingly adopting business document understanding workflows that leverage Vision Language Models (VLMs) for optical character recognition (OCR), given their ability to jointly model layout and language. However, deployment is impeded by data and compute barriers: large enterprises face de-identification pipelines requiring manual validation, while smaller ones lack access to sufficiently large and varied datasets. Synthetic data pipelines that generate millions of <document, OCR> pairs also fall short, as they often fail to capture the nuanced structural and semantic challenges of real-world documents. To address this gap, we introduce **SAVIOR**, a sample-efficient data curation methodology that identifies common failure cases in pretrained VLMs and explicitly curates examples for challenging scenarios such as vertical text, stylized logo text, fine print, and degraded scans. Using SAVIOR, we construct **SAVIOR-TRAIN**, a compact training dataset of 2,234 <document, OCR> tuples, and **SAVIOR-Bench**, a benchmark of 509 financial documents annotated by domain experts. We further introduce **SAVIOR-OCR**, a Qwen2.5-VL-7B-Instruct model fine-tuned on SAVIOR-TRAIN. Experiments show that SAVIOR-OCR achieves a word-level recall of 0.9257 on SAVIOR-Bench, outperforming PaddleOCR 3.0 (0.8685) and Nanonets-OCR-s (0.9040). Beyond recall, we propose **PAIRS**, a structure-aware evaluation metric that quantifies layout fidelity via pairwise spatial relations between tokens; SAVIOR-OCR achieves a PAIRS score of 0.802, demonstrating superior preservation of document structure. To the best of our knowledge, SAVIOR is the first methodology to enable sample-efficient adaptation of VLMs for OCR in enterprise settings, delivering both high accuracy and strong layout fidelity with minimal data and compute.

## 1 Introduction

Modern business document understanding (BDU) workflows Hyperbots Inc. (2025) operate on multi-page, multi-modal documents and execute a sequence of complex downstream tasks such as optical character recognition (OCR), summarization, information extraction, question answering, sentiment analysis, and missing data augmentation. In privacy-sensitive domains such as finance and accounting, enterprises often favor in-house deployment of open-source models such as LLaMA Dubey et al. (2024) and Qwen Yang et al. (2025), rather than relying on externally hosted commercial offerings like GPT-4o Hurst et al. (2024), Gemini Team et al. (2023), or Claude Caruccio et al. (2024). This shift necessitates workflow-level optimizations that balance accuracy, latency, and computational cost within enterprise constraints. One effective architectural choice is to leverage Vision-Language Models (VLMs) solely for perceptual grounding—i.e., extracting structured OCR text from visually complex document images Li et al. (2023a)—followed by delegating reasoning, problem decomposition, and task orchestration to Large Language Models(LLMs) Sapkota et al. (2025); Huang et al. (2024).Under this two-stage design, the deployed BDU workflows must balance high token throughput with a small memory footprint. As shown in Table 1, a 7B model in FP16 precision requires approximately 17 GB of VRAM on an NVIDIA A100 GPU. Quantization techniques such as FP8 or INT4 post-training quantization can reduce VRAM requirements by $2\times$–$4\times$ with minimal accuracy degradation Deepsense.ai (2025). A quantized 7B model can run

Table 1: Approximate GPU VRAM usage (FP16, ≈2K-token context, batch size = 1) during inference on an NVIDIA A100-80GB.

| Model | Params | Approx. VRAM (GB) |
|---|---|---|
| Qwen2.5-VL-7B-Instruct | ≈7B | ≈17 GB Team (2025) |
| LLaMA 3.1 8B-Instruct | ≈8B | ≈16 GB  (blakemart15) |
| LLaMA 3 70B-Instruct | ≈70B | ≈140 GB Team (2024) |

within 8 GB VRAM, allowing up to 10 concurrent replicas per 80 GB A100, which is critical for maximizing GPU utilization under fixed infrastructure budgets. In high-throughput settings, serving heterogeneous requests in parallel is as important as minimizing token-level latency. Inference engines like vLLM Kwon et al. (2023a), which use paged attention and unified KV-cache management, enable near-linear scaling with concurrent requests and support long-context inference (up to 128k tokens OpenAI (2024))—a key requirement in document-heavy workflows. Consider a representative workload: 34,720 invoices per month, at an average of 1.5 pages per invoice. Assuming 4,000 tokens per page (including OCR, summarization, extraction, and reasoning), this results in 208 million tokens per month. Using GPT-4o at a rate of $12.50 per million tokens, processing 208.32 million tokens monthly results in a total cost of approximately $2,604. In contrast, a single 80 GB A100 GPU running multiple quantized 7B model replicas can process up to 59,000 pages per month (at one page per 90 seconds), matching required throughput at comparable cost—while preserving data privacy and providing full control over model behavior. Thus, quantization and inference optimization are not merely performance enhancements; they are prerequisites for deploying scalable, cost-efficient, and privacy-compliant multi-modal BDUs in production.

Even when compute infrastructure is in place, real-world data constraints significantly limit enterprise adoption of two-stage VLM-OCR and LLM workflows. Large enterprises often possess expansive document archives, but using them for model fine-tuning is hindered by extensive de-identification pipelines, with manual validation of personally identifiable information (PII) causing delays of weeks or longer Nanu et al. (2025). On the other hand, small and medium enterprises frequently lack the scale or diversity of labeled documents needed to reliably adapt models to finance-specific contexts, making them heavily reliant on public or synthetic datasets Kumar et al. (2024). While synthetic datasets can be generated at scale, they commonly omit critical layout features and domain-specific semantics—such as vertical text, stylized logos, or noisy scans—and tend to produce "sterile" formats that do not generalize well in commercial deployments Bose et al. (2024). As a result, models trained in such environments often underperform when confronted with real-world variability. These limitations reveal the inadequacy of real or synthetic data volume as a substitute for strategically curated data that reflects failure modes, edge-case semantics, and layout robustness. This also underscores the importance of principled data curation methodologies over brute-force generation strategies, especially for enterprise-grade document understanding systems. To mitigate this gap, we introduce **SAVIOR**, a sample-efficient data curation methodology that explicitly curates data based on known failure modes, reducing both annotation overhead and compute requirements.

Our contributions are as follows:

- SAVIOR, a targeted data curation methodology that identifies and curates examples from common OCR failure scenarios (e.g., vertical text, logo-embedded fields, fine print, noisy scans), enabling sample-efficient OCR alignment.

- Two accompanying resources: SAVIOR-TRAIN, a 2,234-sample curated training set, and SAVIOR-BENCH, a 509-document benchmark annotated by finance-domain experts.

- PAIRS, a novel evaluation metric that measures pairwise relational similarity between predicted and ground-truth word positions, providing the first structure-aware assessment of OCR outputs.

## 2  RELATED WORK

The landscape of document understanding has shifted significantly with the advent of multimodal and layout-aware models. Traditional OCR pipelines such as Tesseract Smith (2007), EasyOCR,

and PP-OCR Du et al. (2020) employ modular architectures with discrete stages for text detection, recognition, and heuristic layout analysis. While these systems are efficient and interpretable, they often exhibit brittle performance under real-world enterprise conditions, including skewed alignments, visual degradation, occlusions (e.g., stamps, logos), and dense multilingual footers Rao et al. (2025); Kaushik et al. (2024). Furthermore, their limited contextual reasoning capabilities hinder performance in tasks like key-value extraction, cross-page entity linking, and document question answering. VLMs have emerged as a more robust alternative, offering unified architectures that jointly model textual and visual context. Early efforts such as LayoutLM Xu et al. (2020) and LayoutLMv3 Huang et al. (2022) introduced 2D positional embeddings to incorporate layout structure, enabling improvements on form-based tasks like FUNSD Jaume et al. (2019) and CORD Park et al. (2019). However, these models remained reliant on external OCR engines and struggled with visual anomalies.

A major shift occurred with encoder-decoder architectures such as Donut Kim et al. (2022), which bypass OCR entirely by generating structured outputs directly from pixels. Donut achieved strong results on semi-structured documents, but its performance declined on visually complex layouts, such as invoices that combine tables with prose. Subsequent models like UDOP Tang et al. (2023) and DocOwl Hu et al. (2024) pushed this direction forward by integrating instruction tuning and enhanced visual-text alignment, enabling end-to-end learning for tasks including document QA and visual question answering (VQA). Simultaneously, general-purpose multimodal systems such as BLIP-2 Li et al. (2023b), LLaVA Liu et al. (2023), and InternVL Chen et al. (2024) achieved strong results on benchmarks like OCRBench v2 Lu et al. (2023) and DocVQA Mathew et al. (2021), demonstrating competitive performance without extensive domain-specific finetuning. In parallel, OCR-to-LLM pipelines have gained prominence, particularly for reasoning-intensive applications. Recent work Wang et al. (2023); Huang et al. (2023) shows that high-quality OCR (e.g., PP-OCR Du et al. (2020)) paired with LLMs such as GPT-4o, Claude 3, or Qwen2.5 can outperform VLMs on downstream tasks involving compositionality and numerical reasoning. This paradigm is especially effective in enterprise contexts, where challenges include tabular consistency, long-range references, and complex metadata interpretation. OCR+LLM pipelines benefit from flexible prompting, structured input formats, and ease of debugging—important considerations in privacy-sensitive and regulated industries.

Despite these advances, both VLM-based and OCR+LLM systems continue to struggle with enterprise-specific edge cases, such as vertical or low-contrast text, rotated headers, logo overlaps, fine-print clauses, and non-standard fonts. Publicly available benchmarks often fail to capture these phenomena, and synthetic corpora such as SynthDoG Kim et al. (2022) and IIT-CDIP Lewis et al. (2006) do not reflect the visual noise and semantic variability present in real-world documents. Privacy constraints further limit the availability of annotated enterprise datasets, making robust generalization across formats and layouts a persistent challenge. On the efficiency front, recent work on quantization Yu et al. (2025); Goyal et al. (2024) and parameter-efficient finetuning (PEFT) Hu et al. (2022); Chang et al. (2024) has improved the deployability of VLMs in resource-constrained environments. Nevertheless, most of these techniques are benchmarked only on standard layouts, and few have been stress-tested against enterprise-specific anomalies. Even state-of-the-art models exhibit hallucination, omission, or token reordering errors in documents with complex visual hierarchies or embedded stamps within tabular regions.

In summary, both VLM-based OCR and OCR+LLM pipelines now represent the state-of-the-art in business document understanding. VLMs offer strong visual-textual alignment and robustness to layout noise, while OCR+LLM pipelines excel at compositional reasoning and structured interpretation. However, both approaches are fundamentally limited by the lack of curated data that captures enterprise-specific failure modes. To address this gap, we introduce **SAVIOR**—a targeted data curation methodology designed to expose VLMs to representative edge cases and enable sample-efficient adaptation. We release **SAVIOR-TRAIN**, a curated dataset of 2,234 training samples emphasizing visually challenging enterprise layouts, and **SAVIOR-Bench**, a 509-sample benchmark of annotated invoices with edge-case artifacts. Our fine-tuned model, **SAVIOR-OCR**, surpasses larger commercial baselines on real-world invoice OCR, while requiring orders of magnitude less training data and memory.

# 3 METHODOLOGY

SAVIOR (**S**ample-efficient **A**daptation of **V**ision models for **I**ntelligent **O**CR **R**epresentation) is a targeted data curation methodology designed to align Vision-Language Models (VLMs) with the structural and semantic requirements of enterprise OCR. The guiding principle is not to maximize dataset size, but to maximize coverage of *high-impact failure modes* with minimal samples. Unlike conventional OCR pipelines that emphasize character-level transcription, SAVIOR explicitly curates training data to preserve semantic meaning, layout structure, and hierarchical relationships required by downstream LLMs in business document workflows.

## 3.1 OVERVIEW OF THE SAVIOR PIPELINE

The methodology proceeds in three stages:

1. **Failure Mode Identification.** We first characterize common breakdowns in pretrained VLMs by running Qwen2.5-VL-7B-Instruct on a held-out validation set of 1,000 invoices and auditing its OCR outputs. Domain experts annotated recurring failure cases, including token fragmentation, layout misordering, and missed fine print. This process yielded a taxonomy of critical scenarios where transcription errors have outsized impact on downstream tasks such as compliance, risk analysis, and structured field extraction.

2. **Targeted Data Curation.** For each identified failure scenario, we curated representative document–OCR pairs. Rather than indiscriminately generating synthetic data, SAVIOR emphasizes *authentic edge cases* sourced from enterprise workflows. To ensure sufficient diversity, actual documents were combined with *systematically generated failure cases* that emulate vertical text, stylized logos, or degraded scans but do not contain sensitive or domain-specific content. This hybrid strategy provides realistic variability while avoiding over-reliance on either raw enterprise data or sterile synthetic corpora.

3. **Balanced Dataset Construction.** Curated examples were balanced across failure modes to form a compact but diverse training corpus. The objective is to ensure sufficient coverage of each mode while keeping annotation overhead low, thereby enabling sample-efficient fine-tuning.

## 3.2 CRITICAL FAILURE SCENARIOS

The following scenarios were explicitly targeted in SAVIOR-TRAIN:

- **Vertical Text Orientation.** Headers and annotations in vertical layouts often cause token fragmentation (e.g., "SUBTOTAL" → "S U B T O T A L").
- **Fine Print and Regulatory Text.** Small-font disclaimers and compliance clauses are frequently missed, leading to downstream regulatory blind spots.
- **Multi-column Layouts.** Incorrect reading order in multi-column structures produces jumbled streams that disrupt logical segmentation.
- **Stylized and Logo-embedded Text.** Key entities within logos or stylized headers are often omitted, producing incomplete vendor or company records.
- **Degraded Image Quality.** Scans with noise, skew, or blur corrupt OCR outputs and propagate errors into structured extraction.
- **Mixed Content Types.** Printed–handwritten combinations (e.g., approvals, forms) challenge VLMs and yield fragmented records.
- **Hierarchical Structure Loss.** Flattened or ignored headers, bullets, and indentation degrade semantic grouping and logical flow.

## 3.3 ALIGNMENT WITH EVALUATION

SAVIOR explicitly curates data to stress both *semantic fidelity* and *layout preservation*. These axes correspond to the evaluation metrics introduced later: word-level recall and layout similarity. By designing SAVIOR-TRAIN around failure modes that most severely disrupt these metrics, the methodology ensures that gains in OCR performance translate directly into improved enterprise workflows.

## 3.4 SAMPLE SIZE JUSTIFICATION

SAVIOR-TRAIN was deliberately capped at 2,234 samples. This constraint demonstrates that strategically curated data, even at small scale, can outperform million-sample synthetic corpora. The methodology therefore enables high OCR accuracy with minimal data and compute, supporting deployment in resource-constrained enterprise environments.

## 4 DATASETS

In this section, we present the datasets introduced as part of the SAVIOR methodology: **SAVIOR-TRAIN**, a curated training set targeting critical OCR failure modes, and **SAVIOR-Bench**, an evaluation benchmark for assessing semantic and structural fidelity in enterprise document understanding. Representative paired examples showing original documents and corresponding SAVIOR-OCR outputs are provided in Figure 2.

### 4.1 SAVIOR-TRAIN

OCR errors that disrupt semantic meaning or structural coherence can cause cascading failures across information extraction, summarization, and compliance tasks. SAVIOR-TRAIN addresses this by selecting and curating samples that directly correspond to such failure modes. The final training set comprises 2,234 document-OCR pairs, strategically balanced across a range of challenging scenarios:

- **Vertical text:** 200 documents (9.0%) containing rotated or vertically aligned text essential for financial field recognition.
- **Fine print:** 100 documents (4.5%) featuring small-font regulatory text critical for compliance extraction.
- **Multi-column layouts:** 150 documents (6.7%) requiring precise reading order to preserve context.
- **Logo-embedded text:** 500 documents (22.4%) with stylized text elements containing entity information.
- **Degraded image quality:** 200 documents (9.0%) exhibiting real-world scanning artifacts such as blur, skew, and compression.
- **Handwritten content:** 50 documents (2.2%) integrating mixed printed and handwritten text.
- **Structured headers:** 1,000 documents (44.8%) preserving formatting hierarchy for logical segmentation.
- **Bold/emphasized text:** 1,034 documents (46.3%) containing visual emphasis markers that aid importance detection.

### 4.2 SAVIOR-BENCH

SAVIOR-Bench is a test dataset comprising 509 documents drawn from real-world business workflows. Each document is annotated to assess not only token-level transcription accuracy but also the preservation of semantic content and structural layout essential for downstream LLM-based processing. The benchmark reflects the distribution of failure modes in SAVIOR-TRAIN, enabling rigorous evaluation of a model's ability to address enterprise-specific challenges such as layout fidelity, semantic segmentation, and regulatory content extraction. Evaluation on SAVIOR-Bench emphasizes the end-to-end impact of OCR quality on downstream tasks, including structured field extraction, compliance analysis, and document classification.

### 4.3 DATA ANNOTATION

All annotations were produced by a specialized team comprising eight finance and accounting graduates with an average age of 23, supervised by a domain expert with 14 years of industry experience. The annotation team, based in Bengaluru, India, focused on capturing semantic intent, visual

hierarchy, and layout fidelity which are criteria aligned with the needs of downstream enterprise applications. A.3 depicts the Inter-Annotator scores on the SAVIOR-TRAIN dataset.

# 5 EVALUATION METRICS

We evaluate models on SAVIOR-Bench and report two complementary metrics designed to assess both semantic completeness and structural fidelity:

## 5.1 WORD-LEVEL RECALL

Word-level recall quantifies the proportion of ground-truth words correctly recovered by the OCR system, counting each occurrence separately. Let $G$ denote the multiset of ground-truth words and $P$ the multiset of predicted words. For each $w \in G$, a match is counted if $w \in P$, and one instance of $w$ is then consumed from $P$. Recall is defined as:

$$\text{Recall} = \frac{1}{|G|} \sum_{w \in G} \mathbf{1}\{\, w \in P \,\},$$

where $\mathbf{1}\{\cdot\}$ is the indicator function.

We adopt word-level recall instead of character-level accuracy, since downstream applications rely on complete tokens. For example, predicting "Totl" instead of "Total" may be close at the character level, but it fails entirely in a key–value extraction task. Word-level recall thus better reflects semantic fidelity in enterprise document workflows.

## 5.2 LAYOUT SIMILARITY METRIC - PAIRS

While existing metrics capture transcription quality, they do not account for whether the spatial structure of the document is preserved. In enterprise workflows, layout fidelity is critical: an output that contains the correct words but misplaces them can render key–value extraction or tabular parsing unusable. For example, if an invoice contains "Total: $1,500" but the text is misplaced near the amount to be paid instead of the discount amount, word-level recall would still rate the transcription as correct. However, downstream LLMs for key–value extraction would misinterpret the field, resulting in incorrect mappings in enterprise workflows. We therefore introduce a layout similarity metric that quantifies structural accuracy.

We represent both ground truth and predictions as 2D arrays. Each document is divided into lines, and within each line, words are placed into horizontal cells. Every word $i$ is assigned coordinates $(x_i, y_i)$, where $x_i$ denotes the index of the cell containing the center character of the word, and $y_i$ is the line number. This representation makes it possible to compare word positions consistently between prediction and ground truth. For each unordered pair $(i, j)$, $1 \leq i < j \leq N$, we compute horizontal, vertical, and Euclidean distances in the ground truth,

$$h_{ij} = |x_j - x_i|, \quad v_{ij} = |y_j - y_i|, \quad d_{ij} = \sqrt{(x_j - x_i)^2 + (y_j - y_i)^2},$$

and analogously $h'_{ij}, v'_{ij}, d'_{ij}$ in the prediction.

The relative error for each pair is

$$e_{ij} = \frac{|d'_{ij} - d_{ij}|}{d_{ij} + \epsilon},$$

which we convert into a bounded similarity score

$$s_{ij} = 1 - e_{ij}^2,$$

where the quadratic penalty ensures that larger deviations are penalized more strongly. The raw layout similarity is then

$$S_{\text{raw}} = \frac{2}{N(N-1)} \sum_{1 \leq i < j \leq N} s_{ij}.$$

To normalize across documents with different scales, we compute $z$-scored distances $z_{ij} = (d_{ij} - \mu)/\sigma$ and $z'_{ij} = (d'_{ij} - \mu)/\sigma$, where $\mu$ and $\sigma$ are the mean and standard deviation of $\{d_{ij}\}$. The normalized similarity is

$$S_z = \frac{2}{N(N-1)} \sum_{1 \leq i < j \leq N} \left( 1 - \left( \frac{|z'_{ij} - z_{ij}|}{|z_{ij}| + \epsilon} \right)^2 \right).$$

This metric directly evaluates whether relative word positions are preserved, complementing word-level recall by capturing structural fidelity in OCR outputs.

## 6 TRAINING STRATEGY AND EXPERIMENTAL SETUP

### 6.1 TRAINING STRATEGY

Our fine-tuning methodology is designed for sample-efficient adaptation by strategically targeting key model components. We apply **Low-Rank Adaptation (LoRA)** with a rank of 32 and scaling factor $\alpha = 64$ to two specific areas: **the final layer of the vision encoder** and **the language model decoder**. This targeted approach for fine-tuning the last encoder layer allows the model to refine its high-level visual feature extraction for enterprise-specific challenges, such as stylized logos and fine print. Simultaneously, strengthening the decoder is crucial for improving its robustness in generating accurate text from degraded or noisy inputs, such as poor-quality scans.

### 6.2 IMPLEMENTATION DETAILS

The vision encoder and language model are optimized with learning rates of $1 \times 10^{-4}$ and $5 \times 10^{-4}$ respectively. Training uses a batch size of 1 with gradient accumulation over 10 steps. All models are trained on 2 NVIDIA H100 80GB GPUs using CUDA 12.3 and driver version 535.216.03. We evaluate models on SAVIOR-Bench, a 509-document benchmark annotated by domain experts. Inference is performed with top-$p = 0.9$ and temperature$= 0.1$.

### 6.3 BASELINE MODELS.

We include commercial and open-source OCR systems such as Nanonets-OCR-s and PaddleOCR 3.0 as baselines. These systems were evaluated in their native inference settings without fine-tuning on SAVIOR-TRAIN. This choice reflects an architectural incompatibility: while SAVIOR-OCR and Qwen models generate sequence-aligned word streams, Nanonets outputs a markup-style representation and PaddleOCR produces bounding-box keyed dictionaries. These formats cannot be straightforwardly aligned to the token-sequence supervision used in SAVIOR-TRAIN without building custom adapters. Furthermore, layout fidelity is only reported for models whose outputs can be controlled through prompting to yield sequential text. As a result, layout similarity metrics are not applicable to Nanonets or PaddleOCR, and we restrict these evaluations to models with prompt-controllable outputs.

## 7 RESULTS AND ANALYSIS

### 7.1 WORD RECALL

As show in Table 2, Qwen2.5VL-7B-Instruct, fine-tuned with SAVIOR, achieves the highest word-level recall of 0.9257 on SAVIOR-BENCH, outperforming both open-source and commercial baselines, including Nanonets-OCR-s (0.9040) and PaddleOCR 3.0 (0.8685). Despite being trained on only 2,234 curated samples, the model demonstrates strong generalization, highlighting the sample-efficiency of the SAVIOR methodology. The 3B variant achieves a recall of 0.9167, while the 32B model reaches 0.9239, offering marginal gains at substantially higher computational cost. This suggests that the 7B model offers the best trade-off between performance and efficiency for practical deployment in enterprise workflows.

Table 2: Performance comparison on SAVIOR-Bench. Layout metric evaluates structure-aware fidelity. All Qwen models were fine-tuned on 2,234 samples using SAVIOR.

| Model | # Params | Word Recall | Layout Metric |
|---|---|---|---|
| Nanonets-OCR-s | ∼3B | 0.903979 | N/A |
| PaddleOCR 3.0 | N/A | 0.868466 | N/A |
| Qwen2.5VL-3B-Instruct | ∼3B | 0.916745 | 0.706812 |
| Qwen2.5VL-7B-Instruct | ∼7B | **0.925748** | 0.706616 |
| Qwen2.5VL-32B-Instruct | ∼32B | 0.923889 | **0.714861** |
| GPT-4.0 | N/A | 0.885481 | 0.582667 |
| Mistral OCR | N/A | 0.861161 | 0.538257 |
| Gemini-2.5-Flash | N/A | 0.914493 | 0.607481 |
| olmOCR-7B-0725-FP8 | ∼7B | 0.746395 | 0.586474 |

Table 3: Average raw layout similarity scores (higher is better).

| Model | Euclidean | Horizontal | Vertical |
|---|---|---|---|
| olmOCR-7B-0725-FP8 | 0.7006 | 0.5756 | 0.7831 |
| Mistral-OCR | 0.6526 | 0.5175 | 0.7456 |
| Qwen2.5VL-7B-Instr. | 0.8200 | 0.6978 | 0.8895 |
| GPT-4o | 0.7140 | 0.5483 | 0.7847 |
| Gemini-2.5-Flash | 0.7560 | 0.5803 | 0.7781 |
| Qwen2.5VL-3B-Instr. | 0.8214 | 0.6940 | **0.8945** |
| Qwen2.5VL-32B-Instr. | **0.8356** | **0.7203** | 0.8761 |

## 7.2 PAIRS METRIC FOR LAYOUT PRESERVATION

Table 3 reports raw layout similarity scores across horizontal, vertical, and Euclidean distances. Qwen2.5VL models achieve the highest scores in all dimensions, with Euclidean similarity around 0.82 and vertical similarity close to 0.89. Gemini-2.5-Flash and GPT-4o achieve intermediate performance, while Mistral OCR shows the lowest alignment quality, particularly along the horizontal axis (0.5175). Across all models, vertical similarity scores consistently exceed horizontal scores, suggesting that baseline alignment is more reliably preserved than intra-line spacing. Table 4 shows layout similarity after z-score normalization, which accounts for scale differences between documents. Relative rankings remain consistent: Qwen models lead across all dimensions, and Mistral OCR continues to lag. Vertical alignment remains more stable than horizontal alignment, even after normalization, possibly due to OCR systems' difficulty in capturing font-size variation and kerning that affect horizontal spacing.

## 7.3 ABLATION STUDY

To assess deployment readiness under resource constraints, we evaluate 4-bit quantized variants of the 3B and 7B models using merged-weight representations. As shown in Table 5, quantization results in modest performance drops: the 7B model's recall decreases by 0.0086, with negligible impact on layout similarity. The 3B model sees a recall drop of 0.0303 but remains competitive. These results indicate that SAVIOR-trained models retain strong OCR capabilities under aggressive compression, supporting their use in low-resource or edge deployment settings.

## 8 CONCLUSION AND FUTURE-WORK

We presented **SAVIOR**, a sample-efficient data curation methodology for aligning Vision Language Models (VLMs) with OCR tasks in enterprise settings. SAVIOR targets known failure modes in pretrained VLMs—such as vertical text, embedded logos, fine print, and degraded scans—through explicit, minimal, and high-utility data curation. We introduced two resources: **SAVIOR-TRAIN**, a 2,234-sample curated training dataset, and **SAVIOR-Bench**, a 509-document benchmark with expert OCR annotations from the finance and accounting domain. We introduced **PAIRS**, a novel evaluation metric that captures pairwise relational similarity between predicted and ground-truth word

Table 4: Average $z$-score normalised layout similarity scores (higher is better).

| Model | Euclidean | Horizontal | Vertical |
|---|---|---|---|
| olmOCR-7B-0725-FP8 | 0.4198 | 0.4340 | 0.6057 |
| Mistral-OCR | 0.3704 | 0.3732 | 0.5702 |
| GPT-4o | 0.4164 | 0.4148 | 0.6178 |
| Gemini-2.5-Flash | 0.4567 | 0.4506 | 0.6231 |
| Qwen2.5-VL-3B-Instr. | 0.5548 | 0.5622 | **0.7140** |
| Qwen2.5VL-7B-Instr. | 0.5557 | 0.5648 | 0.7119 |
| Qwen2.5-VL-32B-Instr. | **0.5747** | **0.5906** | 0.6916 |

Table 5: Performance of 4-bit quantized Qwen models.

| Model | Recall | Layout Metric |
|---|---|---|
| Qwen2.5-VL-7B-Instruct | **0.9171** | **0.7073** |
| Qwen2.5-VL-3B-Instruct | 0.8864 | 0.6843 |

positions, offering the first structure-aware assessment for OCR outputs. Our model, **SAVIOR-OCR**, a fine-tuned Qwen2.5-VL variant, outperforms large-scale OCR baselines like PaddleOCR and Nanonets-OCR-s on SAVIOR-Bench, while using fewer training samples. These results demonstrate that targeted curation, rather than scale alone, is key to achieving high OCR accuracy in enterprise workflows. The results indicate that SAVIOR-trained models remain robust under 4-bit quantization, supporting deployment in low-memory or latency-sensitive environments.Future work includes extending SAVIOR to additional document types (e.g., contracts, receipts), exploring automated identification of OCR failure modes using model introspection signals such as attention heatmaps and confidence scores, and integrating SAVIOR-style training with multimodal instruction tuning for downstream tasks like key-value extraction and summarization. We believe SAVIOR lays the groundwork for lightweight, domain-adaptive, and privacy-conscious document understanding systems for enterprise AI.

### ACKNOWLEDGMENTS

We acknowledge the collaboration with a Fintech startup, conducted under strict data privacy protocols. All invoice data used for annotation were processed internally and remained confidential. No personally identifiable information (PII) was included in training or evaluation, and synthetic data was generated through controlled anonymization. Annotations were carried out by commerce graduates under expert supervision; annotators were fairly compensated and informed about the research objectives. All model development and data usage adhered to ethical standards, with a focus on privacy, fairness, and real-world enterprise applicability.

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

# A  APPENDIX

## A.1  COMPARISON OF QUANTIZATION STRATEGIES

We perform 4-bit and 8-bit quantization on the 3B and 7B LoRA-adapter-merged variants of the Qwen2.5-VL-Instruct model family, and evaluate their throughput when deployed using the vLLM framework  Kwon et al. (2023b) in In Table 6.[1]

Table 6: Memory (VRAM) and throughput (tokens/sec) across quantized models for SAVIOR-OCR variants

| Model | Quantization method | GPU | VRAM (GB) | Throughput (tokens/sec) |
|---|---|---|---|---|
| Qwen2.5-VL-3B-Instruct | 8-bit BNB | A100 | 74.0 | 164.69 |
| | 8-bit BNB | H100 | 56.3 | 343.65 |
| | 4-bit BNB | A100 | 74.3 | 1085.81 |
| | 4-bit BNB | H100 | 66.9 | 2783.16 |
| Qwen2.5-VL-7B-Instruct | 8-bit BNB | A100 | 72.1 | 195.48 |
| | 8-bit BNB | H100 | 56.5 | 342.80 |
| | 4-bit BNB | A100 | 75.6 | 977.60 |
| | **4-bit BNB** | **H100** | **56.4** | **3471.41** |

For both Qwen2.5-VL-3B-Instruct and Qwen2.5-VL-7B-Instruct models, 4-bit quantization yields significantly higher throughput compared to 8-bit, with the 7B model on H100 achieving the highest throughput of 3,471.41 tokens/sec.

## A.2  THROUGHPUT ANALYSIS ON VLLM VS PYTORCH

To assess the deployment viability of our SAVIOR-OCR models in latency-sensitive environments, we measured token-level throughput across two inference engines: vLLM and PyTorch.

As shown in Table 7, vLLM consistently achieves the highest throughput in all models. These experiments demonstrate the importance of selecting optimized inference engines to meet latency and throughput requirements in enterprise settings.

Table 7: Throughput (tokens/sec) across inference engines for SAVIOR-OCR (merged and 4-bit quantized) variants on single 80GB A-100 GPU.

| Inference Engine | OCR Model Variant | Throughput (tokens/sec) |
|---|---|---|
| vLLM | **Qwen2.5-VL-3B-Instruct** | **1085.81** |
| | Qwen2.5-VL7B-Instruct | 977.60 |
| PyTorch | Qwen2.5-VL-3B-Instruct | 424.98 |
| | Qwen2.5-VL-7B-Instruct | 507.61 |

## A.3  INTER-ANNOTATOR AGREEMENT SCORES FOR SAVIOR-TRAIN

To evaluate the consistency of human annotations, we computed inter-annotator agreement (IAA) on the OCR transcriptions produced by two independent annotators per document in Table 8. Since the outputs are free-form strings, we used fuzzy string similarity (normalized Levenshtein ratio) as our agreement metric. Across 187 double-annotated samples, we observed a mean similarity score of 0.761, indicating moderate-to-strong agreement.

---

[1]We observed qualitatively similar quantization behavior for the 32B model but focus our analysis on smaller variants (3B, 7B) given their suitability for efficient deployment in enterprise settings.

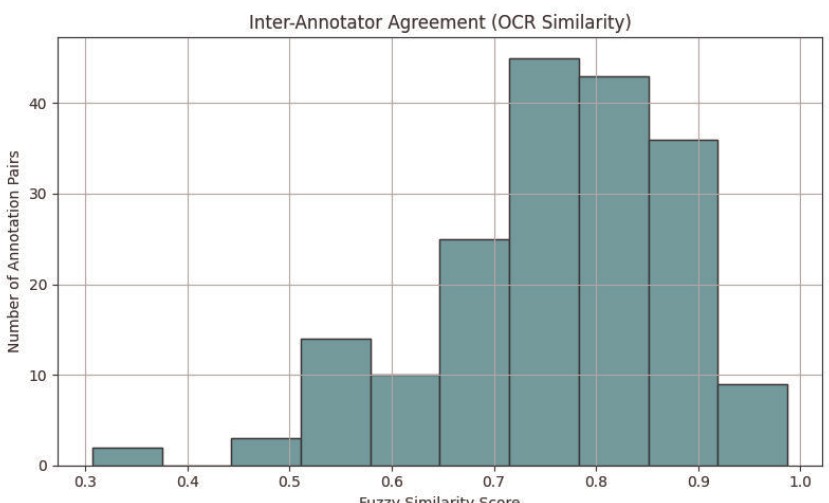

Figure 1: Histogram of inter-annotator similarity scores (OCR annotations).

Table 8: Inter-Annotator Agreement (IAA) on OCR annotations using fuzzy string similarity.

| Metric | Value |
|---|---|
| Number of annotation pairs | 187 |
| Mean similarity | 0.761 |
| Minimum similarity | 0.307 |
| Maximum similarity | 0.987 |
| Standard deviation | 0.117 |

## A.4 DATASET AVAILABILITY

SAVIOR-TRAIN contains 2,234 <document,OCR> tuples and SAVIOR-Bench contains 509 <document,OCR> tuples. Due to enterprise privacy and compliance constraints, we are unable to release the dataset and code. We believe the detailed methodology provides a clear blueprint for other researchers to apply the SAVIOR principles in their respective domains for reproducibility. We are exploring options for a possible release of a de-identified version of the datasets in the future.

## A.5 SAMPLE DOCUMENTS FROM SAVIOR-BENCH

Figure 2 shows representative samples from SAVIOR-BENCH showing original documents (left) and SAVIOR-OCR outputs (right), demonstrating cross-language OCR and failure mode handling.

**APT**

R
E
C
E
I
V
E
R

Olivia Martinez
742 Evergreen Terrace
Springfield, IL 62704
USA

S
U
P
P
L
I
E
R

Greenfield Supplies Ltd.
125 Industrial Park Drive
Suite 400
Dallas, TX 75207
USA

Invoice for "**Lawnmowing Services**" provided from 12-08-2025 to 13-08-2025
at **Evergreen Terrace, Springfield, USA.**

Amount Due: $1400.00
Payment Terms: Net 30

**Terms and Conditions**
By engaging our lawnmowing services, the client agrees to provide clear and
safe access to the property on the scheduled service day. All pets, vehicles, and
personal items must be removed from areas requiring maintenance. Service fees
are due upon completion unless otherwise arranged in writing. The provider is
not liable for damage to concealed objects such as irrigation systems, cables, or
landscaping features that are not clearly marked or disclosed in advance.
Cancellations or rescheduling requests must be made at least 24 hours prior to
the scheduled appointment to avoid a cancellation fee.

*Please pay by due date*

(a) Financial document with stylized logo, bold text, structured headers and handwritten components.

```
<LOGO> APT </LOGO>

                    Olivia Martinez
                    742 Evergreen Terrace
RECEIVER            Springfield, IL 62704
                    USA

                    Greenfield Supplies Ltd.
                    125 Industrial Park Drive
SUPPLIER            Suite 400
                    Dallas, TX 75207
                    USA

Invoice for **"Lawnmowing Services"** provided from 12-08-2025 to 13-08-2025
at **Evergreen Terrace, Springfield, USA**.

Amount Due: $1400.00
Payment Terms: Net 30

#Terms and Conditions
By engaging our lawnmowing services, the client agrees to provide clear and
safe access to the property on the scheduled service day. All pets,
vehicles, and personal items must be removed from areas requiring
maintenance. Service fees are due upon completion unless otherwise arranged
in writing. The provider is not liable for damage to concealed objects such
as irrigation systems, cables, or landscaping features that are not clearly
marked or disclosed in advance. Cancellations or rescheduling requests must
be made at least 24 hours prior to the scheduled appointment to avoid a
cancellation fee.

Please pay by due date
```

SAVIOR-OCR output (a).

**DECLARANDO**

**FACTURA**

Número de factura: 2025-001
Fecha de emisión: 25/09/2025

**Proveedor**
Servicios de Jardinería Verde S.L.
Calle Mayor 123
28001 Madrid, España
NIF: B12345678

**Cliente**
Juan Pérez
Calle Sol 45
28004 Madrid, España
NIF: 12345678A

**Descripción de los servicios**
• Corte de césped (jardín 200 m²) – 1 servicio – 50,00 €
• Recogida y eliminación de residuos vegetales – 1 servicio – 20,00 €

Subtotal: 70,00 €
IVA (21%): 14,70 €

Total a pagar: 84,70 €

Método de pago: Transferencia bancaria

Fecha de vencimiento: 10/10/2025
Cuenta bancaria (IBAN): ES76 2100 0418 4502 0005 1332

(b) Spanish document with stylized logo, structured headers and bold text

```
<LOGO> DECLARANDO </LOGO>

#**FACTURA**

Número de factura: 2025-001
Fecha de emisión: 25/09/2025

**Proveedor**
 Servicios de Jardinería Verde S.L.
 Calle Mayor 123
 28001 Madrid, España
 NIF: B12345678

**Cliente**
 Juan Pérez
 Calle Sol 45
 28004 Madrid, España
 NIF: 12345678A

**Descripción de los servicios**
· Corte de césped (jardín 200 m²) – 1 servicio – 50,00 €
· Recogida y eliminación de residuos vegetales - 1 servicio – 20,00 €

Subtotal: 70,00 €
IVA (21%): 14,70 €

Total a pagar: 84,70 €

Método de pago: Transferencia bancaria

Fecha de vencimiento: 10/10/2025
Cuenta bancaria (IBAN): ES76 2100 0418 4502 0005 1332
```

SAVIOR-OCR for (b).

Figure 2

**CIRCLE**
DESIGN STUDIO

**Invoice no**: 01234
**Issued to**: Olivia Wilson
**Due date**: 30/11/2030

Description of Services

Software Development Services (July 2025) – ₹75,000
Project Consultation (10 hours @ ₹2,000/hr) – ₹20,000
Maintenance & Support – ₹15,000

**Subtotal**: ₹110,000
**GST (18%)**: ₹19,800
**Total Amount Due:** ₹129,800

Bank Name: Wardiere
Account No: 0123 4567 8901
Account Name: Claudia Alves

*This invoice has been approved*

```
<LOGO> The
Circle
DESIGN STUDIO </LOGO>

**Invoice no**: 01234
**Issued to**: Olivia Wilson
**Due date**: 30/11/2030

#Description of Services

Software Development Services (July 2025) – ₹75,000
Project Consultation (10 hours @ ₹2,000/hr) – ₹20,000
Maintenance & Support – ₹15,000

**Subtotal**: ₹110,000
**GST (18%)**: ₹19,800
**Total Amount Due**: ₹129,800

Bank Name: Wardiere
Account No: 0123 4567 8901
Account Name: Claudia Alves

This invoice has been approved
```

(c) Invoice with stylized logo, bold text and hand-written components

SAVIOR-OCR for (c).

The company's consolidated financial statements reflect steady growth in revenue during the fiscal year. Net sales increased by 12.4% compared to the prior year, driven primarily by higher demand in the North American market and favorable foreign exchange rates. Operating income rose accordingly, though at a slower pace, due to increased expenditures on research and development aimed at supporting long-term innovation.

Cash flow from operations remained positive, totaling $58.3 million, which represents a 9% increase over the previous year. The improvement was attributed to stronger collections on receivables and tighter working capital management. However, capital expenditures also grew, as the company invested heavily in upgrading its manufacturing facilities and expanding digital infrastructure to meet evolving customer needs.

All financial information, reports, and statements provided by the Company are intended for informational purposes only and should not be construed as investment, legal, or tax advice. The Company makes no warranties, express or implied, regarding the accuracy, completeness, or reliability of the information provided. Users acknowledge that reliance on such information is at their own risk, and the Company shall not be held liable for any loss or damages arising from the use of the materials.

The company's consolidated financial growth in revenue during the fiscal year. Net sales increased by 12.4% compared to the prior year, driven primarily by higher demand in the North American market and favorable foreign exchange rates. Operating income rose accordingly, though at a slower pace, due to increased expenditures on research and development aimed at supporting long-term innovation.

Cash flow from operations remained positive, totaling $58.3 million, which represents a 9% increase over the previous year. The improvement was attributed to stronger collections on receivables and tighter working capital management. However, capital expenditures also grew, as the company invested heavily in upgrading its manufacturing facilities and expanding digital infrastructure to meet evolving customer needs.

All Financial information, reports, and statements provided by the Company are intended for informational purposes only and should not be construed as investment, legal, or tax advice. The Company makes no warranties, express or implied, regarding the accuracy, completeness, or reliability of the information provided. Users acknowledge that reliance on such information is at their own risk, and the Company shall not be held liable for any loss or damages arising from the use of the materials.

(d) Financial document with multi-column text and fine-print text.

SAVIOR-OCR output for (d).

Figure 2

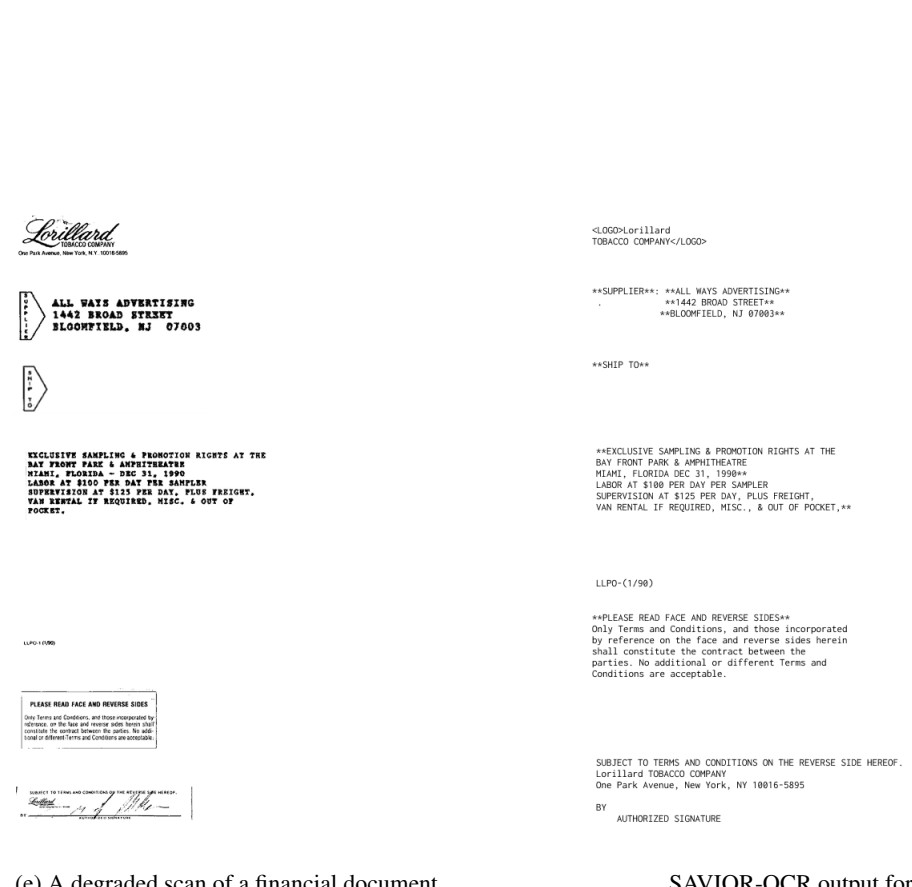

(e) A degraded scan of a financial document.      SAVIOR-OCR output for (e).

Figure 2

