# OpenReview forum: "SAVIOR: Sample-efficient Alignment of Vision-Language Models for OCR Representation"
_ICLR.cc/2026/Conference — ICLR 2026 Conference Withdrawn Submission_

### Official Review · Reviewer_Mk3q · 2025-10-28

**Soundness:** 2
**Presentation:** 2
**Contribution:** 1
**Rating:** 2
**Confidence:** 4

**Summary:**

This paper proposes SAVIOR to enhance the OCR capabilities of MLLMs. The SAVIOR consists of SAVIOR-TRAIN, a training set with 2,234 samples and SAVIOR-Bench, a benchmark with 509 financial documents. Moreover, the authors finetune a Qwen2.5-VL-7B model with SAVIOR-TRAIN, yielding SAVIOR-OCR. However, this paper exhibits major weaknesses in contribution, experiments, and presentation.

**Strengths:**

1. SAVIOR-TRAIN and SAVIOR-Bench are curated based on the eight failure modes of Qwen2.5-VL-7B-Instruct.
2. Several MLLMs, including open-source and proprietary ones, are evaluated on SAVIOR-Bench. The finetuned Qwen2.5-VL-7B-Instruct models achieves the best performance among them.
3. The PAIRS metric is designed to measure layout similarity

**Weaknesses:**

1. The contribution of this paper is limited.
    - There have been many OCR-related benchmarks for MLLMs, e.g., , OmniDocbench, olmOCR-bench, CC-OCR, OCRBench, etc. All of them covers a wide range of scenarios, tasks, and capabilties. However, SAVIOR-Bench only focuses on financial documents and failure modes from Qwen2.5-VL.
    - The failure modes that SAVIOR-TRAIN and SAVIOR-Bench refer to are trivial and have been considered in existing datasets.
    - The size of SAVIOR-TRAIN is much smaller than existing open-source training set (e.g., olmOCR-mix-0225).
2. The experiments are not convincing enough.
    - The MLLMs used for evaluation are not comprehensive enough. Many OCR-oriented MLLMs (dots.OCR, OCRFlux, MonkeyOCR, Dolphin, MinerU2.0, etc) and general MLLMs (InternVL3, LLaMA4, Qwen2.5-VL w/o finetuning) are not evaluation on SAVIOR-Bench.
    - Since the SAVIOR-Bench focuses on limited failure modes of Qwen2.5-VL-7B-Instruct and share the same distribution with SAVIOR-TRAIN, there should be comparsions on other benchmarks like OmniDocBench to verify the effectiveness of SAVIOR-OCR.
    - As specified in line 379, all Qwen models are finetuned with SAVIOR-TRAIN. What about the performance of vanilla Qwen2.5-VL series? How much can finetuning improve the performance?
    - The word recall and PAIRS metrics are not commonly used metrics in the OCR or document parsing field. Moreover, the word recall does not consider the order of words and PAIRS requires location annotations. There have been robust metrics (see OmniDocbench) to evaluate the OCR performance instead of the two metrics adopted in this paper.
3. The presentation of this paper is poor.
    - Some details are missing. For example, what are the prompt and output format of the evaluation on SAVIOR-Bench.
    - The paper title is inappropriate. This work is not relevant to the OCR representation of MLLMs.
    - What does GPT-4.0 refer to in Table 2. Is it the GPT-4o in Table 3?

**Questions:**

Will be the model and datasets publicly released?

---

### Official Review · Reviewer_Lzsk · 2025-10-31

**Soundness:** 1
**Presentation:** 2
**Contribution:** 1
**Rating:** 2
**Confidence:** 5

**Summary:**

This paper describes a 'data curation methodology' and a resulting training and test set of financial documents for OCR. The authors fine-tune (LoRA) QWEN-2.5VL and compare it to the VLMs available through the API (Gemini, GPT, etc) as well as some open-source/commercial OCR models (Nanonets, PaddleOCR). The methodology is simply a selection of challenging OCR use-cases (Vertical text, Fine print, Multi-column layout, Logo-embedded text, degraded image quality, Handwritten content, Structured headers, Bold/emphasized text). The authors propose a layout measurement metric (PAIRS) which measures pairwise displacement between all pairs of words in GT/prediction, and computes the normalized difference between displacements in GT and prediction.

**Strengths:**

The paper highlights some of the common failure cases in document OCR.
The paper is generally easy to follow (though I would suggest to use \citep instead of \cite to make the text more readable and have visual separation between the work name and the citation reference).

**Weaknesses:**

I don't have specific improvement suggestions for this work because I believe it should generally undergo a major revision in order to be accepted to this conference. Below I break down the reasons.

Significance: Low. The paper describes a process of fine-tuning models on a proprietary dataset, which helps improve the metrics on it. The dataset curation methodology lies is selecting challenging usecases. The dataset is proprietary and not available to the public, and therefore neither the methodology nor the models or the dataset would be useful to the wider community.

Quality: Low. The lack of reproducibility is the most significant concern, and the full alignment between the training and test set raises a question about generalization of fine-tuning on this data. The PAIRS metric is not reported for the primary baselines (Nanonets and PaddleOCR). The set of baselines is quite incomplete (I would suggest to include Claude in VLM space, solutions like Google Document AI, Amazon Textract, Microsoft Azure Document Intellignece as commerical solutions, and InternVL, TrOCR, DocOwl as open-source solutions).

Originality: Low. The paper performs parameter-efficient fine-tuning on a collected domain-specific corpus, showing that it outperforms general-purpose solutions. Neither the problem space, nor the solution is novel. The proposed PAIRS metric is basically an implementation of the layout-based metrics (see ex. "The Significance of Reading Order in Document Recognition and Its Evaluation", ICDAR'13).

**Questions:**

None

---

### Official Review · Reviewer_vx7o · 2025-11-02

**Soundness:** 2
**Presentation:** 2
**Contribution:** 2
**Rating:** 2
**Confidence:** 4

**Summary:**

This paper made two contributions
1. It curates a new dataset called SAVIOR-TRAIN with 2k samples for training visual language models for OCR tasks: it contains document image and “OCR” pairs.
2. It also releases a benchmark called SAVIOR-Bench that contains five hundred financial documents and plus a new metric for measuring layout consistency.

The author finetuned Qwen models on this dataset, and reported better performance compared to existing models on the proposed benchmarks.

**Strengths:**

The paper addresses an important problem for efficiently tuning a language model for OCR tasks, which requires high-level of precision. The released dataset could be useful.

**Weaknesses:**

This paper has several important issues:
1. Presentation: I think the overall writing is a bit wordy and there are many confusing parts.
    - For example, the authors use “OCR” a lot in this paper – e.g., “<document, OCR> pairs” – however it is not well defined.
    - Is it a string or some structured representation (e.g., (x,y,w,h,text) for each detected object)?
    - Also I think there are too many engineering details in the paper (e.g., the discussion of the memory cost of the models in the introduction, line 053) – that makes me feel this is somewhat a technical report.
    - For the results in the table – if you’ve fined-tuned the Qwen2.5-VL models, can you use the SAVIOR-OCR name rather than the Qwen models as it’s rather confusing whether it’s the base model or the finetuned ones.
    - Also I have an important question regarding the PAIRS metric, see in the questions section below.
2. Methodology: the primary novelty of the paper seems to be the dataset curation method that selects some special types of documents to be included in the training – however there are two major issues:
    - it’s already known that document recognition is by nature a long tail problem, and including more corner cases is going to be beneficial – the authors don’t address this in the literature; and
    - If the authors want to rigorously test this method, they should also ablate and compare different dataset strategies and validate which method is the best. However this is not reported in the paper, either.
3. Experimental design: to justify the usefulness of the dataset (or the dataset curation method), one should not only test on the in-domain benchmark (i.e., SAVIOR-bench) but also other OCR datasets (e.g., olmoOCR bench). Otherwise, it’s natural to see improved scores as one is training in-domain on the SAVIR-train dataset.

Given all these limitations, I don’t think the issues can be resolved in this review cycle and I think this paper should benefit from another round of revision.

**Questions:**

In the layout similarity metric, the authors mention the “coordinates” of the text:
1. How are the coordinates obtained for the source document words as well as the OCR’d text? Does the model generate the coordinate directly, or do you just render the text and you take it from the rendered piece?
2. And why do you not consider the width and height of the object – since it takes four variables to define a rectangular object in a 2D plane?

---

### Official Review · Reviewer_i9UA · 2025-11-03

**Soundness:** 2
**Presentation:** 3
**Contribution:** 1
**Rating:** 2
**Confidence:** 5

**Summary:**

This paper presents SAVIOR, a sample-efficient data curation method for adapting Vision-Language Models to OCR. SAVIOR identifies common failure cases and builds two datasets: SAVIOR-TRAIN (2,234 samples) and SAVIOR-Bench (509 samples). A fine-tuned Qwen2.5-VL-7B model trained on this data outperforms PaddleOCR and Nanonets on word recall and layout fidelity.

**Strengths:**

- the entire SAVIOR data pipeline, train and test splits, is manually annotated by domain experts.
- the paper is well written, and experimental procedures and results are well explained.

**Weaknesses:**

- The critical failures that motivate SAVIOR (outlined in section 3.2) are similar to the motivation for olmOCR [\(Poznanski et al, 2025\)](https://arxiv.org/abs/2502.18443). The evaluation strategy also shares similarities with olmOCR-Bench. Yet, while SAVIOR is benchmarked against olmOCR, similarities are not acknowledged, nor the olmOCR paper is cited.
- The proposed approach is very specific to the domains SAVIOR is designed against.
- The two metics used both have significant limitations: the word-level recall metric unnecessary penalizes minor stylistic differences (e.g., `$ 1500` vs `$1500`), while the layout similarity enforces a stricter output than necessary. Other works (olmOCR-Bench, [OmniDocBench v 1.5](https://arxiv.org/abs/2412.07626v1)) have shown how output requirements that are too strict end up unfairly penalizing some systems over others.
- The SAVIOR system is impossible to compare with other works: no aspect of the pipeline described in this paper is openly released, and SAVIOR is not evaluated on any open source benchmark.

**Questions:**

N/A

---

### Note · Authors · 2025-12-23

I have read and agree with the venue's withdrawal policy on behalf of myself and my co-authors.